# ACTIVE TUNING

## ABSTRACT

We introduce Active Tuning, a novel paradigm for optimizing the internal dynamics of recurrent neural networks (RNNs) on the fly. In contrast to the conventional sequence-to-sequence mapping scheme, Active Tuning decouples the RNN's recurrent neural activities from the input stream, using the unfolding temporal gradient signal to tune the internal dynamics into the data stream. As a consequence, the model output depends only on its internal hidden dynamics and the closed-loop feedback of its own predictions; its hidden state is continuously adapted by means of the temporal gradient resulting from backpropagating the discrepancy between the signal observations and the model outputs through time. In this way, Active Tuning infers the signal actively but indirectly based on the originally learned temporal patterns, fitting the most plausible hidden state sequence into the observations. We demonstrate the effectiveness of Active Tuning on several time series prediction benchmarks, including multiple super-imposed sine waves, a chaotic double pendulum, and spatiotemporal wave dynamics. Active Tuning consistently improves the robustness, accuracy, and generalization abilities of all evaluated models. Moreover, networks trained for signal prediction and denoising can be successfully applied to a much larger range of noise conditions with the help of Active Tuning. Thus, given a capable time series predictor, Active Tuning enhances its online signal filtering, denoising, and reconstruction abilities without the need for additional training.

## 1 INTRODUCTION

Recurrent neural networks (RNNs) are inherently only robust against noise to a limited extent and they often generate unsuitable predictions when confronted with corrupted or missing data (cf., e.g., Otte et al., 2015). To tackle noise, an explicit noise-aware training procedure can be employed, yielding denoising networks, which are targeted to handle particular noise types and levels. Recurrent oscillators, such as echo state networks (ESNs) (Jaeger, 2001; Koryakin et al., 2012; Otte et al., 2016), when initialized with teacher forcing, however, are highly dependent on a clean and accurate target signal. Given an overly noisy signal, the system is often not able to tune its neural activities into the desired target dynamics at all. Here, we present a method that can be seen as an alternative to regular teacher forcing and, moreover, as a general tool for more robustly tuning and thus synchronizing the dynamics of a generative differentiable temporal forward model—such as a standard RNN, ESN, or LSTM-like RNN (Hochreiter & Schmidhuber, 1997; Otte et al., 2014; Chung et al., 2014; Otte et al., 2016)—into the observed data stream.

The proposed method, which we call Active Tuning, uses gradient back-propagation through time (BPTT) (Werbos, 1990), where the back-propagated gradient signal is used to tune the hidden activities of a neural network instead of adapting its weights. The way we utilize the temporal gradient signal is related to learning parametric biases (Sugita et al., 2011) and applying dynamic context inference (Butz et al., 2019). With Active Tuning, two essential aspects apply: First, during signal inference, the model is not driven by the observations directly, but indirectly via prediction error-inducted temporal gradient information, which is used to infer the hidden state activation sequence that best explains the observed signal. Second, the general stabilization ability of propagating signal hypotheses through the network is exploited, effectively washing out activity components (such as noise) that cannot be modeled with the learned temporal structures within the network. As a result, the vulnerable internal dynamics are kept within a system-consistent activity milieu, effectively decoupling it from noise or other unknown distortions that are present in the defective actual signal.

In this work we show that Active Tuning elicits enhanced signal filtering abilities, without the need for explicitly training distinct models for exactly such purposes. Excitingly, this method allows for instance the successful application of an entirely noise-unaware RNN (trained on clean, ideal data) under highly noisy and unknown conditions.

In the following, we first detail the Active Tuning algorithm. We then evaluate the RNN on three time series benchmarks—multiple superimposed sine waves, a chaotic pendulum, and spatiotemporal wave dynamics. The results confirm that Active Tuning enhances noise robustness in all cases. The mechanism mostly even beats the performance of networks that were explicitly trained to handle a particular noise level. It can also cope with missing data when tuning the predictor's state into the observations. In conclusion, we recommend to employ Active Tuning in all time series prediction cases, when the data is known to be noisy, corrupted, or to contain missing values and the generative differentiable temporal forward model—typically a particular RNN architecture—knows about the potential underlying system dynamics. The resulting data processing system will be able to handle a larger range of noise and corrupted data, filtering the signal, generating more accurate predictions, and thus identifying the underlying data patterns more accurately and reliably.

## 2 ACTIVE TUNING

Starting point for the application of Active Tuning is a trained temporal forward model. This may be, as mentioned earlier, an RNN, but could also be another type of temporal model. The prerequisite is, however, a differentiable model that implements dependencies over time, such that BPTT can be used to reversely route gradient information through the computational forward chain. Without loss of generality, we assume that the model of interest, whose forward function may be referred to as $f_M$, fulfills the following structure:

$$(\boldsymbol{\sigma}^t, \mathbf{x}^t) \xrightarrow{f_M} (\boldsymbol{\sigma}^{t+1}, \tilde{\mathbf{x}}^{t+1}), \tag{1}$$

where $\boldsymbol{\sigma}^t$ is the current latent hidden state of the model (e.g. the hidden outputs of LSTM units, their cell states, or any other latent variable of interest) and $\mathbf{x}^t$ is the current signal observation. Based on this information $f_M$ generates a prediction for the next input $\tilde{\mathbf{x}}^{t+1}$ and updates its latent state $\boldsymbol{\sigma}^{t+1}$ accordingly.

Following the conventional inference scheme, we feed a given sequence time step by time step into the network and receive a one-time step ahead prediction after each particular step. Over time, this effectively synchronizes the network with the observed signal. Once the network dynamics are initialized, which is typically realized by teacher forcing, the network can generate prediction and its dynamics can be driven further into the future in a closed-loop manner, whereby the network feeds itself with its own predictions. To realize next time step- and closed-loop predictions, direct contact with the signal is inevitable to drive the teacher forcing process. In contrast, Active Tuning decouples the network from the direct influence of the signal. Instead, the model is permanently kept in closed-loop mode, which initially prevents the network from generating meaningful predictions. Over a certain time frame, Active Tuning keeps track of the recent signal history, the recent hidden states of the model, as well as its recent predictions. We call this time frame (retrospective) tuning horizon or tuning length (denoted with $R$).

The principle of Active Tuning can best be explained with the help of Figure 1 and Algorithm 1. The latter gives a more formal perspective onto the principle. Note that for every invocation of the procedure a previously unrolled forward chain (from the previous invocation or an initial unrolling) is assumed. $\mathcal{L}$ refers to the prediction error of the entire unrolled prediction sequence and the respective observations, whereas $\mathcal{L}^{t'}$ refers to the local prediction error just for a time step $t'$. With every new perceived and potentially noise-affected signal observation $\mathbf{x}^t$, one or multiple tuning cycles are performed. Every tuning cycle hereby consists of the following stages: First, from the currently believed sequence of signal predictions (which is in turn based on a sequence of hidden states) and the actual observed recent inputs, a prediction error is calculated and propagated back into the past reversely along the unfolded forward computation sequence. The temporal gradient travels to the very left of the tuning horizon and is finally projected onto the seed hidden state $\boldsymbol{\sigma}^{t-R}$, which is then adapted by applying the gradient signal in order to minimize the encountered prediction error. This adaption can be done using any gradient-based optimizer. Note that in this paper, we exclusively use Adam (Kingma & Ba, 2015), but other optimizers are possible as well. Second, after the adap-

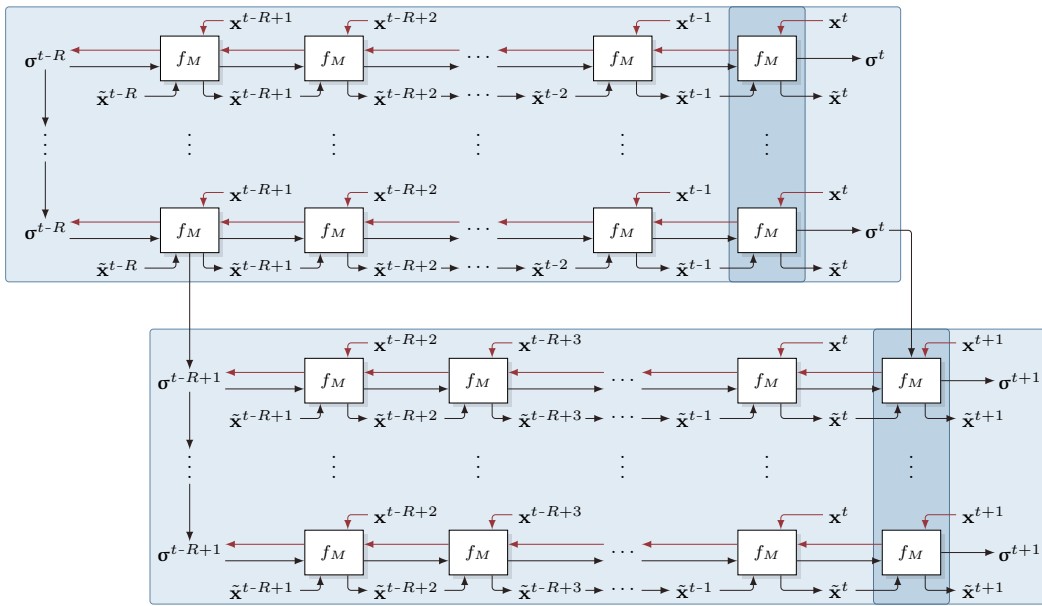

Figure 1: Illustration of Active Tuning principle over two consecutive world time steps. $f_M$ refers to the forward pass of the base model. $\mathbf{x}^t, \mathbf{x}^{t-1}$ etc. are the recent potentially defective signal observations, whereas $\tilde{\mathbf{x}}^t, \tilde{\mathbf{x}}^{t-1}$ etc. are the respective predictions (outputs of the model's forward function $f_M$). $R$ denotes the length of the retrospective tuning horizon, that is, the number of time steps the prediction error is projected into the past using BPTT. $\boldsymbol{\sigma}^{t-R}$ refers to the latent (hidden) state of $M$ at the beginning of the tuning horizon, which essentially seeds the unfolding prediction sequence (black lines). $\boldsymbol{\sigma}^{t-R}$ is actively optimized based on the back-projected prediction error gradient (red lines).

---

**Algorithm 1:** Active Tuning procedure

> **Input** : Current observation $\mathbf{x}^t$
> **Output:** Prediction $\tilde{\mathbf{x}}^t$ (filtered output), predictive hidden state $\boldsymbol{\sigma}^t$
>
> $\tilde{\mathbf{x}}^t, \boldsymbol{\sigma}^t \leftarrow f_M(\tilde{\mathbf{x}}^{t-1}, \boldsymbol{\sigma}^{t-1})$    /* *Generate current prediction based on previous forward chain* */
>
> **for** $c \leftarrow 1$ **to** $C$ **do**    /* *Perform multiple tuning cycles* */
>> **for** $t' \leftarrow t$ **to** $t - R$ **do**    /* *Back-propagate prediction error* */
>>
>> $$\mathbf{g}^{t'} \leftarrow \frac{\partial \mathcal{L}}{\partial \boldsymbol{\sigma}^{t'}} = \frac{\partial \mathcal{L}^{t'}}{\partial \boldsymbol{\sigma}^{t'}} + \begin{cases} \mathbf{g}^{t'+1} \dfrac{\partial \boldsymbol{\sigma}^{t'+1}}{\partial \boldsymbol{\sigma}^{t'}} & \text{if } t' < t \\ 0 & \text{otherwise} \end{cases}$$
>>
>> **end for**
>>
>> $\boldsymbol{\sigma}^{t-R} \leftarrow \text{update}(\boldsymbol{\sigma}^{t-R}, \mathbf{g}^{t-R})$    /* *Perform tuning step (e.g. with Adam update rule)* */
>>
>> **for** $t' \leftarrow t - R + 1$ **to** $t$ **do**    /* *Roll out forward chain again based on adapted hidden state* */
>>> $\tilde{\mathbf{x}}^{t'}, \boldsymbol{\sigma}^{t'} \leftarrow f_M(\tilde{\mathbf{x}}^{t'-1}, \boldsymbol{\sigma}^{t'-1})$
>>
>> **end for**
>
> **end for**
>
> **return** $\tilde{\mathbf{x}}^t$, $\boldsymbol{\sigma}^t$

---

tion of this seed state (and maybe the seed input as well) the prediction sequence is rolled out from the past into the present again, effectively refining the output sequence towards a better explanation of the recently observed signal. Each tuning cycle thus updates the current prediction $\tilde{\mathbf{x}}^t$ and the current hidden state $\boldsymbol{\sigma}^t$ from which a closed-loop future prediction can be rolled out, if desired. To transition into the next world time step, one forward step has to be computed. The formerly leftmost seed states can be discarded and the recorded history is shifted by one time step, making $\boldsymbol{\sigma}^{t-R+1}$ the new seed state that will be tuned within the next world time step. From then on, the procedure is

repeated, yielding the continuous adaptive tuning process. As a result, the model is predominantly driven by its own imagination, that is, its own top down predictions. Meanwhile, the predictions themselves are adapted by means of the temporal gradients based on the accumulated prediction error, but not by the signal directly. In a nutshell, Active Tuning realizes a gradient-based mini-optimization procedure on any of the model's latent variables within one world time step. While it needs to be acknowledged that this process draws on additional computational resources, in this paper we investigate the resulting gain in signal processing robustness.

Intuitively speaking, Active Tuning tries to fit known temporal patterns, as memorized within the forward model, to the concurrently observed data. Due to the strong pressure towards consistency maintenance, which is naturally enforced by means of the temporal gradient information in combination with the repeatedly performed forward passes of the hidden state activities, the network will generate adaptations and potential recombinations of patterns that it has learned during training. Occurrences that cannot be generated from the repertoire of neural dynamics will therefore not appear (or only in significantly suppressed form) in the model's output. As a consequence, there is a much smaller need to strive for noise robustness during training. Our results below indeed confirm that the model may be trained on clean, idealized target signals. However, imprinting a slight denoising tendency during training proves to be useful when facing more noisy data. Enhanced with our Active Tuning scheme, the model will be able to robustly produce high-quality outputs even under extremely adverse conditions—as long as (some of) the assumed target signals are actually present. Our scheme is thus a tool that can be highly useful in various application scenarios for signal reconstruction and flexible denoising. Nevertheless, it should be mentioned that with Active Tuning the computational overhead for inference scales with the number of tuning cycles and the tuning length.

## 3  EXPERIMENTS

In order to investigate the abilities of Active Tuning we studied its behavior at considering three different types of time series data, namely, one-dimensional linear dynamics, two-dimensional non-linear dynamics, and distributed spatiotemporal dynamics. For all three problem domains we used a comparable setup except for the particular recurrent neural network architectures applied. We trained the networks as one time step ahead predictors whose task is to predict the next input given both the current input and the history of inputs aggregated in the latent hidden state of the models. The target sequences were generated directly from the clean input sequences by realizing a shift of one time step. Moreover, we trained networks under six different denoising conditions (normally distributed) per experiment, where we fed a potentially noisy signal into the network and provide the true signal (one time step ahead) as the target value (Lu et al., 2013; Otte et al., 2015; Goodfellow et al., 2016). These conditions are determined by their relative noise ratios: $0.0$ (no noise), $0.05$, $0.1$, $0.2$, $0.5$, and $1.0$, where the ratios depend on the respective base signal statistics. For instance, a noise ratio of $0.1$ means that the noise added to the input has a standard deviation of $0.1$ times the standard deviation of the base signal. As a result we obtained predictive *denoising experts* for each of these conditions. All models were trained with Adam (Kingma & Ba, 2015) using its default parameters (learning rate $\eta = 0.001$, $\beta_1 = 0.9$ and $\beta_2 = 0.999$) over 100 (first two experiments) or 200 (third experiment) epochs, respectively.

### 3.1  MULTI-SUPERIMPOSED OSCILLATOR

The first experiment is a variant of the *multiple superimposed oscillator* (MSO) benchmark (Schmidhuber et al., 2007; Koryakin et al., 2012; Otte et al., 2016). Multiple sine waves with different frequencies, phase-shifts, and amplitudes are superimposed into one signal (cf. Eq. 2 in the Section A.1), where $n$ gives the number of superimposed waves, $f_i$ the frequency, $a_i$ the amplitude, and $\varphi_i$ the phase-shift of each particular wave, respectively. Typically, the task of this benchmark is to predict the further progression of the signal given some initial data points (e.g. the first 100 time steps) of the sequence. The resulting dynamics are comparably simple as they can, in principle, be learned with a linear model. It is, however, surprisingly difficult for BPTT-based models, namely LSTM-like RNNs, to precisely continue a given sequence for more than a few time steps (Otte et al., 2019). For this experiment we considered the $\text{MSO}_5$ dynamics with the default frequencies $f_1 = 0.2$, $f_2 = 0.311$, $f_3 = 0.42$, $f_4 = 0.51$, and $f_5 = 0.63$. An illustration of an exemplary the ground truth signal can be found in Figure 2.

For training, we generated $10\,000$ examples with 400 time steps each, using random amplitudes $a_i \sim [0, 1]$ and random phase-shifts $\varphi_i \sim [0, 2\pi]$. For testing, another $1\,000$ examples were generated. As base model, we used an LSTM network with one input, 32 hidden units, one linear output neuron, and no biases, resulting in $4\,256$ parameters. Additionally, to contrast our results with another state-of-the-art sequence-to-sequence model, temporal convolution networks (TCNs) (Kalchbrenner et al., 2016) were trained. Preliminary experiments showed that seven layers with 1, 8, 12, 16, 12, 8, and 1 feature maps, a kernel size of 3, and standard temporal dilation rate—yielding a temporal horizon of 64 time steps—tended to generate the best performance with a comparable number of parameters (i.e. $4\,682$). Code was taken from Bai et al. (2018).

## 3.2 CHAOTIC PENDULUM

The second experiment is based on the simulation of a chaotic double pendulum. As illustrated in Figure 5 (cf. Section A.2), the double pendulum consists of two joints whose angles are specified by $\theta_1$ and $\theta_2$ and two rods of length $l_1$ and $l_2$. Besides the length of the rods, the masses $m_1$ and $m_2$ affect the behavior of the pendulum. The pendulum's end-effector (where $m_2$ is attached) generates smooth, but highly non-linear trajectories. More precisely, it exhibits chaotic behavior, meaning that already slight changes of the current system state can quickly cause major changes of the pendulum's state over time (Korsch et al., 2008; Pathak et al., 2018). It is thus typically difficult to precisely predict the dynamics of such a system for more than a few time steps into the future, making it a challenging benchmark problem for our purposes.

In the literature, the double pendulum's dynamics are typically described using the equations of motion, given by Eq. 3 and Eq. 4 (cf. ), respectively, which are derived from the Lagrangian of the system and the Euler-Lagrange equations; see Korsch et al. (2008) for details. For simulating the double pendulum, we applied the fourth-order Runge-Kutta (RK4) (Press, 2007) method to numerically integrate the equations of motion. All four parameters $l_1$, $l_2$, $m_1$, and $m_2$ were set to 1.0. A temporal step size of $h = 0.01$ was chosen for numerical integration. The initial state of the pendulum is described by its two angles, which were selected randomly for each sample to be within $\theta_1 \sim [90°, 270°]$ and $\theta_2 \sim [\theta_1 \pm 30°]$ to ensure sufficient energy in the system. One out of ten sequences was initiated with zero angle momenta, that is $\dot{\theta}_1, \dot{\theta}_2 = 0.0$. The number of train and test samples, as well as the sequence lengths were chosen analogously to experiment one. As base model we used an LSTM network with two inputs, 32 hidden units, two linear output neurons, and again no biases. Again, we trained TCNs on this data by changing the number of input and output feature maps to two. Otherwise, the settings were identical to the ones used in experiment one.

## 3.3 SPATIOTEMPORAL WAVE DYNAMICS

In the third experiment we considered a more complex spatiotemporal wave propagation process, based the wave dynamics formalized by Eq. 5 (cf. Section A.3). Here, $x$ and $y$ correspond to a spatial position in the simulated field, while $t$ denotes the time step and $c = 3.0$ the propagation speed factor of the waves. The temporal and spatial approximation step sizes were set to $h_t = 0.1$ and $h_x = h_y = 1.0$, respectively. No explicit boundary condition was applied, resulting in the waves being reflected at the borders and the overall energy staying constant over time.

We generated sequences for a regular grid of $16 \times 16$ pixels. See Figure 4 or Figure 6 for illustrations of the two-dimensional wave. In contrast to the previous two experiments, 200 samples with a sequence length of 80 were generated for training, whereas 20 samples over 400 time steps were used for evaluation. As base network we used a distributed graph-RNN called DISTANA (Karlbauer et al., 2020), which is essentially a mesh of the same RNN module (an LSTM, which consists here of four units only), which is distributed over the spatial dimensions of the problem space (here a two-dimensional grid), where neighboring modules are laterally connected. We chose this wave benchmark, and this recurrent graph network in particular, to demonstrate the effectiveness of Active Tuning in a setup of higher complexity. Moreover, we again trained TCNs, here with three layers, having 1, 8, and 1 feature maps, respectively, using $3 \times 3$ spatial kernel sizes and standard dilation rates for the temporal dimension. Noteworthy, while the applied DISTANA model counts 200 parameters, the applied TCN has an order of magnitude more parameters ($2\,306$). Less parameters yielded a significant drop in performance.

## 4 RESULTS AND DISCUSSION

The quantitative evaluations are based on the root mean square error (RMSE) between the network outputs and the ground truth. All reported values are averages over ten independently randomly initialized and trained models. In order to elaborate on the applicability of each denoising expert on unseen noise conditions, we evaluated all models using the noise ratios 0.0, 0.1, 0.2, 0.5, and 1.0, resulting in 25 baseline scores for each experiment. These baselines were compared on all noise ratios against eight Active Tuning setups, which were based on models trained without any noise (0.0) or with only a small portion of input noise (0.05). The individual parameters of Active Tuning used to produce the results are reported in Section A.5 of the appendix. Note that in all experiments, the latent hidden outputs of the LSTM units (not the cell states) were chosen as optimization target for the Active Tuning algorithm. Furthermore, these hidden states were initialized normally distributed with standard deviation 0.1 in all cases, whereas the cell state were initialized with zero.

### 4.1 MSO RESULTS

The results of the MSO experiments are summarized in Table 1. Active Tuning improves the results for the weakest model (0.0) in all cases (column 3 vs. 7), partially almost by an order of magnitude. Noteworthy, for the inference noise ratio 0.1, driven with Active Tuning the noise-unaware model becomes better than the actual expert. Recall that the model was no retrained, only the paradigm how the model is applied was changed. On the other hand, there is no advantage for Active Tuning when the base network encountered minimal noise (0.05) during training in this experiment. For comparison, a noise-uninformed TCN (0.0) performs better than the respective RNN (cf. column 2 vs. column 3 in Table 1). Active Tuning reverses this disparity. On this benchmark, however, denoising expert TCNs clearly outperform the expert RNNs (cf. Table 6 in Section A.4).

Table 1: MSO noise suppression results (RMSE)

| Inference (signal noise) | TCN | Regular inference RNN | | | | | Active Tuning | |
|---|---|---|---|---|---|---|---|---|
| | 0.0 | 0.0 | 0.1 | 0.2 | 0.5 | 1.0 | 0.0 | 0.05 |
| 0.0 | 0.0411 | **0.0039** | 0.0498 | 0.0781 | 0.1526 | 0.2383 | — | — |
| 0.1 | 0.1341 | 0.5880 | 0.0966 | 0.0993 | 0.1579 | 0.2397 | **0.0912** | 0.0947 |
| 0.2 | 0.2580 | 1.0336 | 0.1734 | **0.1454** | 0.1728 | 0.2446 | 0.1682 | 0.1550 |
| 0.5 | 0.6189 | 1.8713 | 0.4265 | 0.3190 | **0.2538** | 0.2751 | 0.3583 | 0.3611 |
| 1.0 | 1.1676 | 2.6241 | 0.9502 | 0.6437 | 0.4396 | **0.3639** | 0.5699 | 0.6101 |

Table 2: MSO missing data results (RMSE)

| | Missing data probability | | | | | | | | |
|---|---|---|---|---|---|---|---|---|---|
| | 0.1 | 0.2 | 0.3 | 0.4 | 0.5 | 0.6 | 0.7 | 0.8 | 0.9 |
| Regular inference | 0.1206 | 1.0357 | 2.2410 | 2.9601 | 3.3488 | 3.7070 | 4.0039 | 4.0065 | 3.4861 |
| Active Tuning | **0.0321** | **0.0381** | **0.0581** | **0.0826** | **0.1462** | **0.3874** | **0.6209** | **1.0757** | **1.4064** |

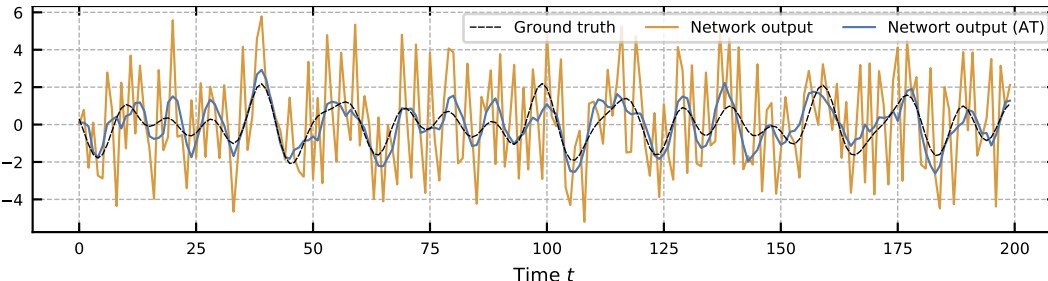

Figure 2: Visual comparison of regular inference (orange) vs. Active Tuning (light blue) on an MSO example with strong noise (noise ratio 1.0) using a noise-unaware LSTM.

To get an impression of the actual improvement of the output quality, consider Figure 2. The noise-unaware model (0.0) produces poor predictions when confronted with strong signal noise (1.0). When driven with Active Tuning instead of regular inference (teacher forcing), the output of the same model becomes smooth and approximates the ground truth reasonably well. Active Tuning thus helps to catch most of the trend information while mostly ignoring noise.

As an additional evaluation, Table 2 demonstrates the ability of Active Tuning to cope with missing data. The results are based on the noise-unaware model. While tuning into the signal, particular observation are missing (dropped out) with a certain probability ranging from 0.1 to 0.9. In case of a missing observation, the prediction of the model is used instead. Already with a dropout chance of 20 % the RNN struggles to tune its hidden neural states into the signal, thus generating an error larger than 1. In contrast, exactly the same RNN model remains significantly more stable when driven with Active Tuning. Even with a dropout chance of $50 - 60\%$ the RNN still produces errors clearly below the reference error of approximately 0.9, which is the RMSE error generated when always predicting zero. Note that with regular inference, the error decreases slightly with the highest dropout rate. This is the case because here the network receives so few inputs such that it starts to produce zero outputs for some sequences.

It seems that during regular inference the network dynamics are overly driven by the input data. When parts of the input is missing, the internal dynamics do not synchronize with the true dynamics because multiple consecutive time steps of consistent signal observations appear necessary. Active Tuning enables the network to reflect on the recent past including its own prediction history. While it attempts to maintain consistency with the learned dynamics, it infers a hidden state sequence that best explains the (even sparsely) encountered observations and thus effectively fills the input gaps retrospectively with predictions that seem maximally plausible.

## 4.2 Pendulum Results

For the pendulum experiment, the potential of Active Tuning becomes even more evident. The results presented in Table 3 indicate that for all noise ratios Active Tuning outperforms the respective expert RNNs, particularly when applied to the model that was trained on small noise (0.05). With increasing noise level, the problem becomes progressively impossible to learn. For example, the 1.0-expert-model does not seem to provide any reasonable function at all, indicated by the worse RMSE score compared to other models (1.0 inference noise row). In contrast, Active Tuning can still handle these extremely unfavorable conditions surprisingly well. Figure 3 shows an exemplary case. The unknown ground truth is plotted against the noisy observations (shown in the left image). The center image shows the prediction of the reference LSTM (trained with 0.05 noise) when regular inference is applied. It is clearly difficult to recognize a coherent trajectory reflecting the dynamics of the double pendulum. With Active Tuning, on the other hand, the same network produces a mostly clean prediction that is relatively close to the ground truth sequence.

Analogously to MSO, we evaluated the robustness against missing data on the pendulum. The results are reported in Table 4. In contrast to the previous scenario (cf. Table 2) it is noticeable that the base model is intrinsically more stable in this experiment. Still, Active Tuning yields a significant improvement in all cases. For the mid-range dropout rates, it decreases the prediction error by approximately an order of magnitude. Even with a dropout rate of 80 % still somewhat accurate predictions are generated. Please note again that the comparison here uses exactly the same RNN (the same structure as well as the same weights) for both regular inference and Active Tuning.

Table 3: Pendulum noise suppression results (RMSE)

| Inference (signal noise) | TCN | Regular inference RNN | | | | | Active Tuning | |
| --- | --- | --- | --- | --- | --- | --- | --- | --- |
| | 0.0 | 0.0 | 0.1 | 0.2 | 0.5 | 1.0 | 0.0 | 0.05 |
| 0.0 | 0.0135 | **0.0091** | 0.1475 | 0.2471 | 0.4459 | 0.5700 | — | — |
| 0.1 | 0.1155 | 0.2097 | 0.1537 | 0.2489 | 0.4463 | 0.5702 | 0.0880 | **0.0865** |
| 0.2 | 0.2272 | 0.4021 | 0.1711 | 0.2545 | 0.4474 | 0.5710 | 0.1423 | **0.1284** |
| 0.5 | 0.5572 | 0.8458 | 0.2702 | 0.2945 | 0.4563 | 0.5764 | 0.2954 | **0.2460** |
| 1.0 | 1.0959 | 1.2753 | 0.5308 | 0.4444 | 0.4918 | 0.5954 | 0.4868 | **0.4030** |

Table 4: Pendulum missing data results (RMSE)

| | Missing data probability | | | | | | | | |
|---|---|---|---|---|---|---|---|---|---|
| | 0.1 | 0.2 | 0.3 | 0.4 | 0.5 | 0.6 | 0.7 | 0.8 | 0.9 |
| Regular inference | 0.0158 | 0.0377 | 0.1493 | 0.3286 | 0.6100 | 0.9370 | 1.2831 | 1.6068 | 1.8457 |
| Active Tuning | **0.0111** | **0.0148** | **0.0190** | **0.0278** | **0.0518** | **0.0977** | **0.1698** | **0.3206** | **0.7932** |

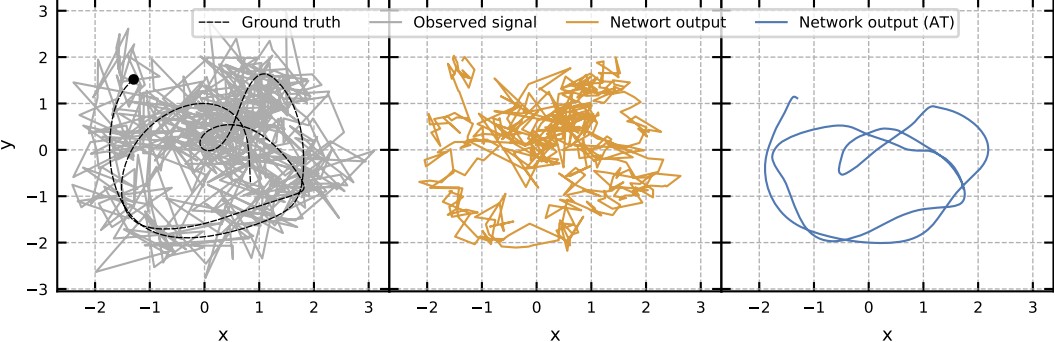

Figure 3: Exemplary comparison of regular inference (orange) vs. Active Tuning (light blue) on the double pendulum's end-effector trajectory; the black dot denotes the start position. Here, the second strongest noise condition (0.5) is shown, using a 0.05-noise LSTM for inference and Active Tuning.

## 4.3 WAVE RESULTS

The results of the wave experiment (Table 5) consistently support the findings from the pendulum experiments. When driven with Active Tuning, the considered models produce better results than the explicitly trained denoising experts on all noise levels. Figure 4 shows in accordance with the previous experiments that the noisy signal observations (1.0) is filtered effectively and latency-free exclusively when using Active Tuning, yielding a smooth signal prediction across the entire spatiotemporal sequence. While the two-dimensional output of the network operating in conventional inference mode is hardly recognizable as a wave, the network output of the same model combined with Active Tuning clearly reveals the two-dimensional wave structure with hardly perceivable deviations from the ground truth. More qualitative results can be found in Figure 6 (Section A.4).

We furthermore compared performance with a common, non-recurrent, sequence-to-sequence learning architecture. Here we considered a standard TCN architecture (Bai et al., 2018), which we also trained to focus on all considered denoising levels. The full performance table is shown in Table 8 (Section A.4). The first result column in Table 5 shows that even the best TCN performance is always outperformed by Active Tuning. Importantly, DISTANA with Active Tuning outperforms the best TCN results on all noise levels even when the DISTANA model was not trained for denoising.

We also performed experiments with other noise distributions (e.g. salt-and-pepper noise). Somewhat surprisingly this manipulation affected the quality of the output only marginally. Thus, in contrast to deep convolutional networks (Geirhos et al., 2018), the denoising RNNs applied here did not overfit to the noise type.

Table 5: Wave noise suppression results (RMSE)

| Inference (signal noise) | TCN (best) | Training (signal noise) | | | | | | |
|---|---|---|---|---|---|---|---|---|
| | | Regular inference RNN | | | | | Active Tuning | |
| | | 0.0 | 0.1 | 0.2 | 0.5 | 1.0 | 0.0 | 0.05 |
| 0.0 | 0.0051 | **0.0007** | 0.0021 | 0.0042 | 0.0096 | 0.0175 | — | — |
| 0.1 | 0.0138 | 0.0268 | 0.0073 | 0.0064 | 0.0100 | 0.0176 | 0.0073 | **0.0062** |
| 0.2 | 0.0252 | 0.0533 | 0.0142 | 0.0106 | 0.0113 | 0.0178 | 0.0097 | **0.0087** |
| 0.5 | 0.0581 | 0.1295 | 0.0362 | 0.0262 | 0.0180 | 0.0197 | 0.0173 | **0.0150** |
| 1.0 | 0.1133 | 0.2368 | 0.0784 | 0.2467 | 0.0345 | 0.0261 | 0.0283 | **0.0213** |

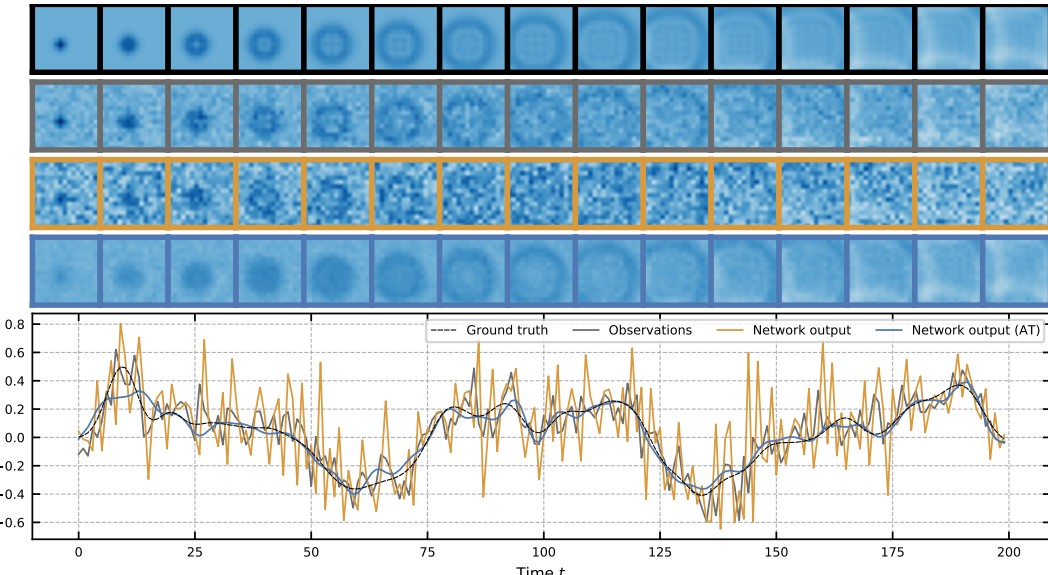

Figure 4: Exemplary comparison of regular inference (orange) vs. Active Tuning (light blue) on wave examples with strong noise (1.0) using DISTANA trained without noise. The four top rows visualize ground truth, noisy observations (network input), network output without, and network output with Active Tuning. The plot below shows the wave activities at the center position.

# 5 CONCLUSION

In this work we augmented RNN architectures with Active Tuning, which decouples the internal dynamics of an RNN from the data stream. Instead of relying on the input signal to set the internal network states, Active Tuning retrospectively projects the dynamic loss signal onto its internal latent states, effectively tuning them. We have shown that RNNs driven with Active Tuning can reliably denoise various types of time series dynamics, mostly yielding higher accuracy than specifically trained denoising expert RNNs. In all cases, however, the augmentation with Active Tuning has beaten the reference RNN with teacher forcing. Moreover, we have shown that Active Tuning increases the tolerance against missing data by a large extent allowing the models to generate accurate prediction even if more than 50 % of the input are missing. Comparisons with TCN have shown that Active Tuning yields superior performance. In the wave experiments, TCNs are consistently outperformed by the similarly trained recurrent graph neural network DISTANA (Karlbauer et al., 2020). When adding Active Tuning, even the noise uniformed DISTANA version outperformed the best TCN networks. Note that even though we used Active Tuning exclusively for RNNs in this paper, it is in general not restricted to such models. We are particularly interested in adapting the principle to other sequence learning models such as TCNs.

While the presented results are all very encouraging, it should be noted that in our experience Active Tuning is slightly slower to tune the network into a clean signal. Seeing that Active Tuning can in principle be mixed with traditional teacher forcing, we are currently exploring switching teacher forcing on and off in an adaptive manner depending on the present signal conditions.

Another concern lies in the applied tuning length and number of tuning cycles. In the presented experiments, we used up to 16 time steps with partially up to 30 tuning cycles. Additional ongoing research aims at reducing the resulting computational overhead. Ideally, Active Tuning will work reliably with a single update cycle over a tuning length of a very few time steps, which would allow to perform Active Tuning along with the regular forward pass of the model in a fused computation step. Additionally, we aim at applying Active Tuning to real-world denoising and forecasting challenges, including speech recognition and weather forecasting.

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

# A APPENDIX

## A.1 MSO EXPERIMENT

The MSO experiment were based on the following equation:

$$\text{MSO}_n(t) = \sum_{i=1}^{n} a_i \sin(f_i t + \varphi_i) \tag{2}$$

## A.2 CHAOTIC PENDULUM EXPERIMENT

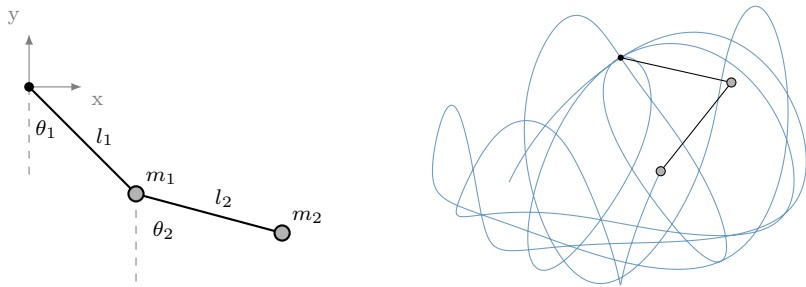

Figure 5: The double pendulum used for data generation of the second experiment and a resulting nonlinear trajectory of the pendulum's end-effector.

The pendulum experiments were based on the following equations:

$$\ddot{\theta}_1 = \frac{\mu g_1 \sin(\theta_2)\cos(\theta_2 - \theta_1) + \mu\dot{\theta}_1^2 \sin(\theta_2 - \theta_1)\cos(\theta_2 - \theta_1) - g_1\sin(\theta_1) + \frac{\mu}{\lambda}\dot{\theta}_2^2\sin(\theta_2 - \theta_1)}{1 - \mu\cos^2(\theta_2 - \theta_1)} \tag{3}$$

$$\ddot{\theta}_2 = \frac{g_2\sin(\theta_1)\cos(\theta_2 - \theta_1) - \mu\dot{\theta}_2^2\sin(\theta_2 - \theta_1)\cos(\theta_2 - \theta_1) - g_2\sin(\theta_2) - \lambda\dot{\theta}_1^2\sin(\theta_2 - \theta_1)}{1 - \mu\cos^2(\theta_2 - \theta_1)}, \tag{4}$$

where

$$\lambda = \frac{l_1}{l_2}, \qquad g_1 = \frac{g}{l_1}, \qquad g_2 = \frac{g}{l_2}, \qquad \mu = \frac{m_2}{m_1 + m_2},$$

and $g = 9.81$ being the gravitational constant.

### A.3 WAVE DYNAMICS EXPERIMENT

The wave experiments were based on the following equation:

$$\frac{\partial^2 u}{\partial t^2} = c^2 \left( \frac{\partial^2 u}{\partial x^2} + \frac{\partial^2 u}{\partial y^2} \right). \tag{5}$$

This equation was solved numerically using the method of second order central difference, yielding

$$u(x, y, t + h_t) \approx c^2 h_t^2 \left( \frac{\partial^2 u}{\partial x^2} + \frac{\partial^2 u}{\partial y^2} \right) + 2u(x, y, t) - u(x, y, t - h_t) \tag{6}$$

with, after solving $\partial^2 u / \partial x^2$ and analogously $\partial^2 u / \partial y^2$ via the same method,

$$\frac{\partial^2 u}{\partial x^2} = \frac{u(x + h_x, y, t) - 2u(x, y, t) + u(x - h_x, y, t)}{h_x^2}, \tag{7}$$

$$\frac{\partial^2 u}{\partial y^2} = \frac{u(x, y + h_y, t) - 2u(x, y, t) + u(x, y - h_y, t)}{h_y^2}. \tag{8}$$

### A.4 FURTHER RESULTS

The performance of the temporal convolution networks (TCNs) at predicting and denoising the MSO, pendulum and spatiotemporal wave dynamics are reported in the following tables (Table 6, Table 7 and Table 8). An additional qualitative evaluation of the wave benchmark on a larger grid shown in Figure 6.

Table 6: TCN MSO noise suppression results (RMSE)

| Inference (signal noise) | Training (signal noise) | | | | |
|---|---|---|---|---|---|
| | 0.0 | 0.1 | 0.2 | 0.5 | 1.0 |
| 0.0 | 0.0411 | **0.0352** | 0.0633 | 0.0968 | 0.1255 |
| 0.1 | 0.1341 | 0.0936 | **0.0789** | 0.1016 | 0.1289 |
| 0.2 | 0.2580 | 0.1764 | 0.1206 | **0.1151** | 0.1386 |
| 0.5 | 0.6189 | 0.4367 | 0.3239 | **0.1898** | 0.1921 |
| 1.0 | 1.1676 | 0.8744 | 0.7431 | 0.3926 | **0.3140** |

Table 7: TCN pendulum noise suppression results (RMSE)

| Inference (signal noise) | Training (signal noise) | | | | |
|---|---|---|---|---|---|
| | 0.0 | 0.1 | 0.2 | 0.5 | 1.0 |
| 0.0 | **0.0135** | 0.0367 | 0.0637 | 0.1369 | 0.2281 |
| 0.1 | 0.1155 | **0.0759** | 0.0817 | 0.1416 | 0.2298 |
| 0.2 | 0.2272 | 0.1407 | **0.1214** | 0.1543 | 0.2339 |
| 0.5 | 0.5572 | 0.3733 | 0.2952 | **0.2278** | 0.2624 |
| 1.0 | 1.0959 | 0.8193 | 0.6748 | 0.4144 | **0.3491** |

Table 8: TCN wave noise suppression results (RMSE)

| Inference (signal noise) | Training (signal noise) | | | | |
|---|---|---|---|---|---|
| | 0.0 | 0.1 | 0.2 | 0.5 | 1.0 |
| 0.0 | 0.0057 | **0.0051** | 0.0064 | 0.0109 | 0.0173 |
| 0.1 | 0.0168 | 0.0143 | **0.0138** | 0.0158 | 0.0207 |
| 0.2 | 0.0321 | 0.0271 | **0.0252** | 0.0253 | 0.0284 |
| 0.5 | 0.0790 | 0.0666 | 0.0611 | **0.0581** | 0.0587 |
| 1.0 | 0.1551 | 0.1320 | 0.1216 | 0.1146 | **0.1133** |

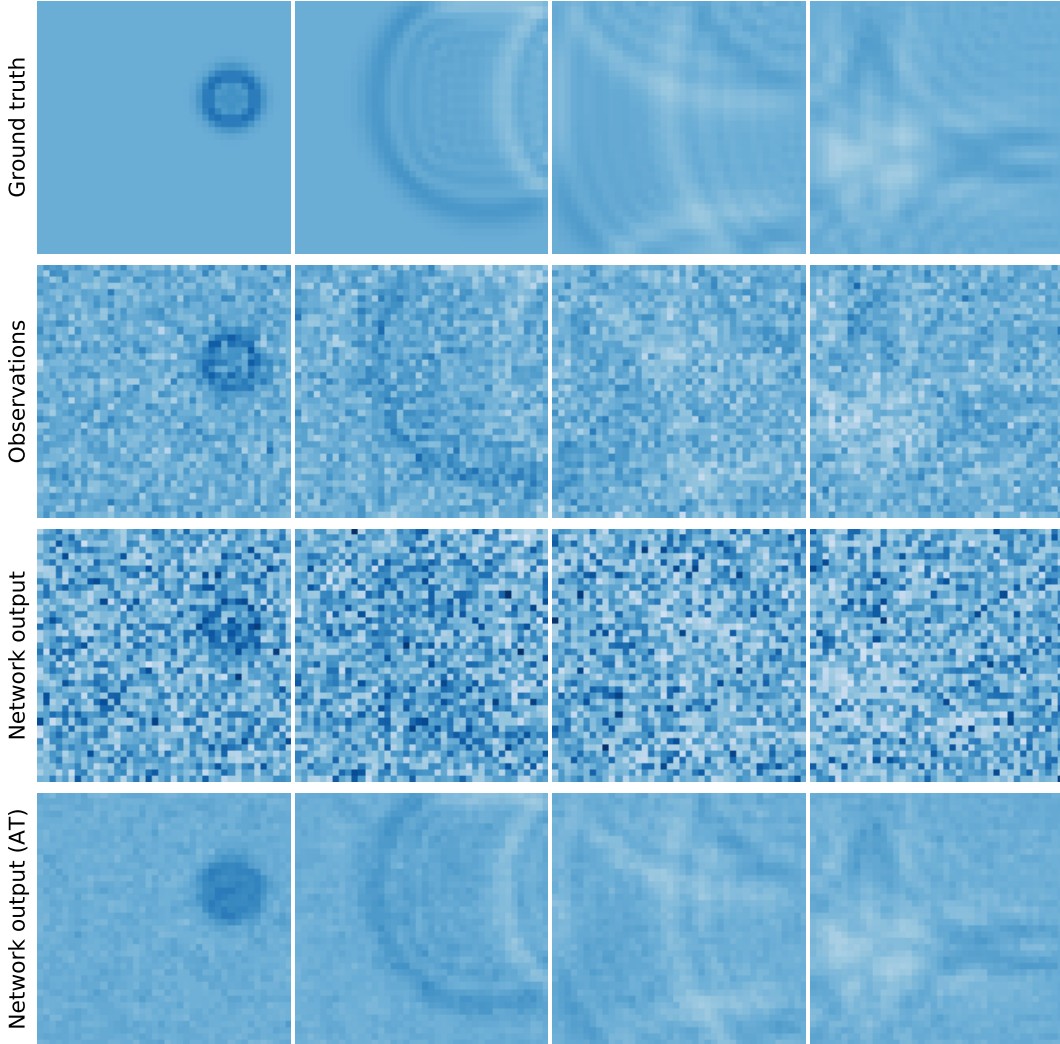

Figure 6: Four consecutive snapshots (from left to right, with distance of 50 time steps ) of the wave propagating through a large $40 \times 40$ grid comparing regular inference and Active Tuning on a noise-unaware network (0.0).

## A.5 ACTIVE TUNING PARAMETERS

The tables below report all parameters of Active Tuning including the parameters for state adaptation with Adam for all experiments.

Table 9: Active Tuning parameters – MSO noise suppression experiment

| Training noise | Signal noise | Tuning length ($R$) | Tuning cycles ($C$) | $\eta$ | $\beta_1$ | $\beta_2$ |
|---|---|---|---|---|---|---|
| 0.0 | 0.1 | 8 | 10 | 0.005 | 0.9 | 0.99 |
| 0.0 | 0.2 | 8 | 10 | 0.005 | 0.9 | 0.99 |
| 0.0 | 0.5 | 14 | 10 | 0.006 | 0.9 | 0.99 |
| 0.0 | 1.0 | 16 | 10 | 0.004 | 0.5 | 0.99 |
| 0.05 | 0.1 | 8 | 10 | 0.008 | 0.9 | 0.99 |
| 0.05 | 0.2 | 8 | 12 | 0.005 | 0.5 | 0.999 |
| 0.05 | 0.5 | 14 | 10 | 0.007 | 0.9 | 0.99 |
| 0.05 | 1.0 | 16 | 10 | 0.006 | 0.5 | 0.9 |

Table 10: Active Tuning parameters – MSO missing data experiment

| Missing data probability | Tuning length ($R$) | Tuning cycles ($C$) | $\eta$ | $\beta_1$ | $\beta_2$ |
|---|---|---|---|---|---|
| $0.1 - 0.5$ | 5 | 20 | 0.005 | 0.9 | 0.99 |
| $0.6 - 0.9$ | 10 | 10 | 0.005 | 0.9 | 0.99 |

Table 11: Active Tuning parameters – pendulum noise suppression experiment

| Training noise | Signal noise | Tuning length ($R$) | Tuning cycles ($C$) | $\eta$ | $\beta_1$ | $\beta_2$ |
|---|---|---|---|---|---|---|
| 0.0 | 0.1 | 8 | 10 | 0.005 | 0.9 | 0.99 |
| 0.0 | 0.2 | 8 | 10 | 0.005 | 0.9 | 0.99 |
| 0.0 | 0.5 | 8 | 10 | 0.004 | 0.5 | 0.99 |
| 0.0 | 1.0 | 12 | 10 | 0.004 | 0.5 | 0.9 |
| 0.05 | 0.1 | 8 | 10 | 0.008 | 0.9 | 0.99 |
| 0.05 | 0.2 | 8 | 10 | 0.005 | 0.5 | 0.99 |
| 0.05 | 0.5 | 8 | 10 | 0.004 | 0.5 | 0.99 |
| 0.05 | 1.0 | 12 | 10 | 0.005 | 0.5 | 0.9 |

Table 12: Active Tuning parameters – pendulum missing data experiment

| Missing data probability | Tuning length ($R$) | Tuning cycles ($C$) | $\eta$ | $\beta_1$ | $\beta_2$ |
|---|---|---|---|---|---|
| $0.1 - 0.6$ | 5 | 20 | 0.005 | 0.9 | 0.99 |
| $0.7 - 0.9$ | 8 | 20 | 0.005 | 0.9 | 0.99 |

Table 13: Active Tuning parameters – wave noise suppression experiment

| Training noise | Signal noise | Tuning length ($R$) | Tuning cycles ($C$) | $\eta$ | $\beta_1$ | $\beta_2$ |
|---|---|---|---|---|---|---|
| 0.0 | 0.1 | 7 | 10 | 0.01 | 0.9 | 0.999 |
| 0.0 | 0.2 | 5 | 17 | $6 \times 10^{-5}$ | 0.0 | 0.999 |
| 0.0 | 0.5 | 4 | 20 | $8 \times 10^{-5}$ | 0.0 | 0.999 |
| 0.0 | 1.0 | 7 | 30 | $4 \times 10^{-5}$ | 0.0 | 0.999 |
| 0.05 | 0.1 | 8 | 12 | 0.012 | 0.9 | 0.999 |
| 0.05 | 0.2 | 5 | 17 | $1 \times 10^{-4}$ | 0.0 | 0.999 |
| 0.05 | 0.5 | 4 | 20 | $1 \times 10^{-4}$ | 0.0 | 0.999 |
| 0.05 | 1.0 | 7 | 30 | $5 \times 10^{-5}$ | 0.0 | 0.999 |

