# OpenReview forum: "Active Tuning"
_ICLR.cc/2021/Conference — Reject_

### Official Review · AnonReviewer4 · 2020-10-28
**An interesting approach in optimizing the internal dynamics of recurrent neural networks**

**Rating:** 8
**Confidence:** 3

**Review:**

This is an interesting paper on an idea introduced by the authors as active tuning. This paper is well-written and clearly explains the proposed active tuning scheme. I read the paper carefully multiple times, and feel that a few inclusions will help the readers better understand the proposed method.

At the base level, this paper builds on optimizing the internal dynamics of a recurrent neural network unlike optimizing internal weights in traditional sequence-to-sequence mapping. This is achieved by decoupling the recurrent neural activities from the input temporal signal and propagating the error (the difference between the estimated input value and the observed input value of the input signal) to tune the internal dynamics of the network. To demonstrate the effectiveness of active tuning the authors trained a distributed graph recurrent neural network (DISTANA) on three datasets with increasing complexity. Datasets included: multiple super-imposed sine waves, a chaotic double pendulum, and spatiotemporal wave dynamics.  On average ten independent experiments were performed and the effectiveness of active tuning was evaluated using root mean square (RMS).  Samples for the experiments were generated using five different noise ratios between 0 and 1 to measure the effectiveness of the proposed method for noisy data scenarios. The network was also trained on no noise to 0.05 noise induced into training data to see if it would help the models better generalize. The results as depicted in graphs show that active tuning is not only robust but generalizes well on noisy data.

Recommendations:
1. The active Tuning algorithm itself is missing from this paper. Even though the explanation is clear, it would help the readers to see the algorithm itself for better understanding. The reviewer referred to Hidden Latent State Inference in a Spatio-Temporal Generative by karlbauer et. al. 2020 (arXiv:2009.09823) for the algorithm.

2. The authors confirm that 10000 and 1000 samples have been generated for all the problem domains tested. However, it is not clear if steps were in place to make sure that no bias was introduced during this sample generation.

3. While the tuning length and tuning cycles were fixed for all three datasets, it is important to see how these values can be optimized based on the complexity of the time series data. Experimental results using a range of values for tunning length and tuning cycles would be beneficial.

---

> ### Author Response · Authors · 2020-11-18
> **Response to Reviewer 3 (4)**
>
> Dear Reviewer 3,
>
> Thank you very much for your positive feedback and your recommendations. Please note that we applied DISTANA only for predicting the Wave Dynamics. In the other two problem cases, standard LSTMs were applied, underlining the general applicability of Active Tuning.
>
> - Yes, we clearly see your demand for an algorithmic description as justified. We added Algorithm 1 including some further explanations in the text and hope this complements the method section satisfactorily.
> - We are not entirely sure about what you mean with bias concerning the data generation. But, of course, we tried to produced datasets that were as unbiased as possible, randomizing the parameters of the generation process accordingly.
> - We actually did vary the tuning length and tuning cycles across the three different problem domains and noise levels (albeit not very much). The particular choices can be found in Table 6, 7, and 8 in Section A.2 of the appendix.

---

### Official Review · AnonReviewer2 · 2020-10-28
**A novel method to tune autoregressive model via hidden state optimization**

**Rating:** 3
**Confidence:** 4

**Review:**

Paper proposes a way to adapt an autoregressive model (RNN in examples) to the incoming noisy signal to generate noise-free data output. The approach is interesting due to applying updates to the hidden state of the past observation. The proposed approached is named Active Tuning and evaluated on 3 toy tasks. The idea sounds interesting, however the lack of comparisons with other approaches and theoretical justification of why this approach is superior makes it hard to convince reader.

Quality: Paper is well written and most of the concepts is clear. However, paper will benefit from a better explanation of the method, simpler diagram and equations to remove uncertainty on implementation.

Originality: I believe the idea is novel and interesting for community. It has a potential to outperform meta-learning and sequence-to-sequence models on the task of model adaptation to noisy samples.

Pros:
- Idea is interesting and has potential.
- Explanation is clear, but still can be improved.
- Provided experiments show benefits of the proposed method with respect to direct regression task (same model trained with less or more noise amount)

Cons:
- Comparison with other techniques such as meta-learning, sequence-to-sequence models is required to understand the potential of the method.
- Same comparisons might be interesting for tuning weights instead of hidden states. Or having only a small part of the model to be tuned (like the last layer).
- Application to more practical problems could benefit the paper. For example image denosing task could be relevant (works like Noise2Noise, Noise2Self etc)

---

> ### Author Response · Authors · 2020-11-18
> **Response to Reviewer 2**
>
> Dear Reviewer 2
>
> Thank you for your constructive feedback on our work.
>
> - By including an algorithmic explanation of our method (see Algorithm 1 in the revised paper), we hope to both increase the comprehensibility of our paradigm's description and remove any uncertainties on implementation details.
> - Your statement about the potential of Active Tuning outperforming other models raised our curiosity. Hence, we incorporated an experiment with a state-of-the-art sequence-to-sequence model, namely, a temporal convolution network (TCN). Indeed, as reported in the modified Table 4 (and Table 5 in the appendix) of our revised paper, we can verify your expectations and have consistently observed that the TCN is outperformed by Active Tuning. However, we would like to again highlight the fact that we did not aim at beating the most sophisticated state-of-the-art ANN in particular domains. Rather, Active Tuning has the potential to generally enhance the performance of prediction models, and particularly recurrent temporal prediction models, without further training when facing missing values and noisy data.
> - "Same comparisons might be interesting for tuning weights instead of hidden states.". This is clearly a very interesting idea, which we hope to elaborate on in future work. What we investigated so far is tuning the cell states, the "hidden states" (unit outputs), and even the signal itself. What we found is that it did not really make a huge difference, which of the mentioned parts are optimized. Typically, tuning just the hidden units outputs works best, but only with a small margin (at least for LSTMs). When tuning the weights catastrophic forgetting might become an issue, which Active Tuning fully avoids, because the model parameters (i.e. the ANN weights) are not modified.

---

### Official Review · AnonReviewer1 · 2020-10-28
**paper is well-written and clear. There are no related work discussed.**

**Rating:** 5
**Confidence:** 3

**Review:**

This paper introduces a propagation method to estimate RNN dynamic parameters during the learning process. The algorithm is introduced well and the paper is clearly written.
The paper misses a related work section on other tuning methods or absence there of under special circumstances.
For the same reason, I am not convinced on the extent of comparisons in the simulation results. A large amount of the focus of the experiments is on the robustness to noise which is fine if there was an equal amount of comparisons against other tuning methods. Otherwise, if the focus of the paper is supposed to be only on noise robustness, I think the motivation in abstract and introduction needs to be clearer.
While the motivation of the paper can be to some extent taken from the results, the introduction does not substantially motivate the problem.
Lastly, I think that majority of details on pages 4 and 5 are unnecessary. Instead, I think a more detailed discussion on comparing additional computational cost of active tuning to other traditional methods would be very useful.

---

> ### Author Response · Authors · 2020-11-18
> **Response to Reviewer 1**
>
> Dear Reviewer 1,
>
> Thank you very much for your review and your suggestions.
>
> - It is very important to emphasize that within this paper Active Tuning is not applied during or for training (it is nonetheless a good idea to incorporate it during training). Instead, the hidden states of pre-trained models are optimized based on the prediction error-induced gradient information. The outputs of the RNNs are thus only driven by the continuously applied gradient-based tuning of the hidden states. Sorry, if we did not make this clear enough. We hope that with inclusion of Algorithm 1 this becomes more comprehensible.
>
> - To the best of our knowledge, there are no comparable approaches to exclusively tune the RNN's hidden states for inference purposes. The main statement that we tried to make with our paper is, however, that Active Tuning can dramatically improve an already existing model (without additional learning) and unfold robustness properties that have not been addressed during training. In order to further interpret the potential of the method, we incorporated results of temporal convolution networks (see updated Table 4 in subsection 4.3. as well as the new Table 5 in the appendix).
> - We appreciate and agree on your point that we exclusively focused on noise filtering and noise robustness. To underline the potential of Active Tuning further, we now performed an additional experiment to demonstrate superior robustness to missing values in time series data when using Active Tuning (see Table 2 of the revised paper).
> - Indeed, the potential computational overhead of using Active Tuning can not be neglected. Yet, this overhead, caused by a gradient-based mini optimization procedure within every global time step, scales with the number of tuning cycles C and tuning horizon R (both essentially depend on the problem, as can be seen in the Tables 6, 7 and 8 of the appendix). We are currently working on reducing the required numbers of C and R, which would significantly reduce the computational overhead. Nevertheless, since we admit the importance of this aspect, we have added a corresponding amendment to Section 2.

---

### Author Response · Authors · 2020-11-18
**General response to the reviews and list of major modifications**

Thank you to all three reviewers for insightful comments, the criticism, and the suggestions. We tried to address the mentioned requests and suggestions and hope all of you will find the paper even more appealing now.

Besides some general minor text reformulation and cosmetics, the major additions in the uploaded revision are as follows:

- A formal algorithmic description (Algorithm 1) with additional explanations.
- Another evaluation that demonstrates that Active Tuning can also handle missing data (Table 2).
- We added a comparison to the performance of a temporal convolution network (TCN) on the spatiotemporal wave dynamics benchmarks (updated Table 4 and new Table 5 in the appendix, TCN setup cf. end of Section 3; results discussed in subsection 4.3 Wave Results).

We thus hope that we can convince also Reviewer 1 and Reviewer 2 to rate our paper above threshold after all (currently rating level 5 and 3, respectively; Reviewer 3 gave rating level 8).

---

### Author Response · Authors · 2020-11-23
**On Revision 23 Nov 2020**

To complete the comparison with TCN and missing data values, we have just uploaded another paper revision.

This revision includes now TCN results for all three problem domains considered. While TCN partially outperforms the respective RNNs when trained on similar denoising levels, Active Tuning applied to a noise-unaware RNN outperforms the noise-unaware TCN. In the conclusions we now also mention the potential to add Active Tuning to TCNs and other feed-forward ANNs.

Additionally, we also added the dropout experimental results to the pendulum data.

Moreover, for the wave experiments, we added another illustrative visualization of the superior performance of Active Tuning in the DISTANA network.

In the light of these new result additions, we have also adapted the conclusions slightly further.

Thank you to all three reviewers for taking the time to reconsider our paper and your consideration and time in general.

Sincerely yours,
the authors.

---

### Decision · Program_Chairs · 2021-01-07
**Final Decision**

**Decision:**

Reject

**Comment:**

This paper introduces a method to estimate dynamics parameters in recurrent structured models during the learning process. All three reviewers agreed that the idea is interesting and the proposed method could be potentially useful. However, two of the three reviewers have a serious concern about the lack of comparison with other approaches. I agree with these two reviewers; due to the lack of discussion and comparison with existing studies, I cannot recommend accepting this submission in its current form.